# TAGExplainer: Narrating Graph Explanations for Text-Attributed Graph Learning Models

## Abstract

Representation learning of Text-Attributed Graphs (TAGs) has garnered significant attention due to its applications in various domains, including recommendation systems and social networks. Despite advancements in TAG learning methodologies, challenges remain in explainability due to the black-box nature of existing TAG representation learning models. This paper presents TAGExplainer, the first method designed to generate natural language explanations for TAG learning. TAGExplainer employs a generative language model that maps input-output pairs to explanations reflecting the model's decision-making process. To address the lack of annotated ground truth explanations in real-world scenarios, we propose first generating pseudo-labels that capture the model's decisions from saliency-based explanations, then the pseudo-label generator is iteratively trained based on three training objectives focusing on faithfulness and brevity via Expert Iteration, to improve the quality of generated pseudo-labels. The high-quality pseudo-labels are finally utilized to train an end-to-end explanation generator model. Extensive experiments are conducted to demonstrate the effectiveness of TAGExplainer in producing faithful and concise natural language explanations. Our code is available at `https://anonymous.4open.science/r/TAGExplainer-6FEB/`.

## 1 Introduction

Graph representation learning, which aims to understand graph-structured data, has garnered significant attention in research, particularly through methods like Graph Neural Networks (GNNs) (Kipf & Welling, 2016; Hamilton et al., 2017). Recently, learning on Text-Attributed Graphs (TAGs), a type of graph where nodes are associated with texts, has received increasing focus from the community due to its universal discriminative power in various domains Zhou et al. (2019); Yang et al. (2021). For example, in e-commerce graphs of recommendation systems, products can be associated with textual descriptions; in social networks, each user can also be enriched by textual descriptions. There have been significant efforts made for TAG learning, including cascaded models which use pre-trained language models to extract text embeddings from nodes and then feed to GNN as node features (Zhou et al., 2019; Zhu et al., 2021; Zhao et al., 2022), and nested models which use one model to jointly process text and graph information (Yang et al., 2021; Chien et al., 2021; Jin et al., 2023). The recent raising of Large Language Models (LLMs) also brings light to LLMs-augmented TAG learning pipelines, such as (He et al., 2023; Zhang et al., 2024). However, similar to graph learning, TAG learning also faces challenges in terms of explainability due to the inherent black-box nature of models. Although LLMs can also generate rationales for explaining their predictions, they do not necessarily reflect the inner decision process of the model (Agarwal et al., 2024; Parcalabescu & Frank, 2024). In this work, we dive into the problem of explainability in TAG learning, which is a rising yet not well-explored area.

For graph learning, explainability methods have received a large amount of attention (Ying et al., 2019; Vu & Thai, 2020; Luo et al., 2020). Existing explanation models often provide feature importance-based explanations, by giving node- and edge-level importance scores, but they lack the ability to explain the semantic information in TAGs since the node- and edge-level importance scores cannot include any information in the text features. Some methods (Ying et al., 2019; Štrumbelj &

Kononenko, 2014; Bach et al., 2015) can also explain the importance of each individual node feature, and thus can generate better TAG explanations by further giving the importance scores of each token. However, TAG predictions are often made based on a subgraph with many nodes and their associated texts, simply giving each token an importance score may bring too much redundant information without a well-integrated and narrated context, causing a low human understandability in generated explanations, as illustrated in Fig. 1 (a). Therefore, a form of more human-understandable explanation is necessary for TAG learning, which should be summative and concise, as one example shown in Fig. 1 (b). The limitation and need for human understandability necessitates our research on natural language explanations for TAGs.

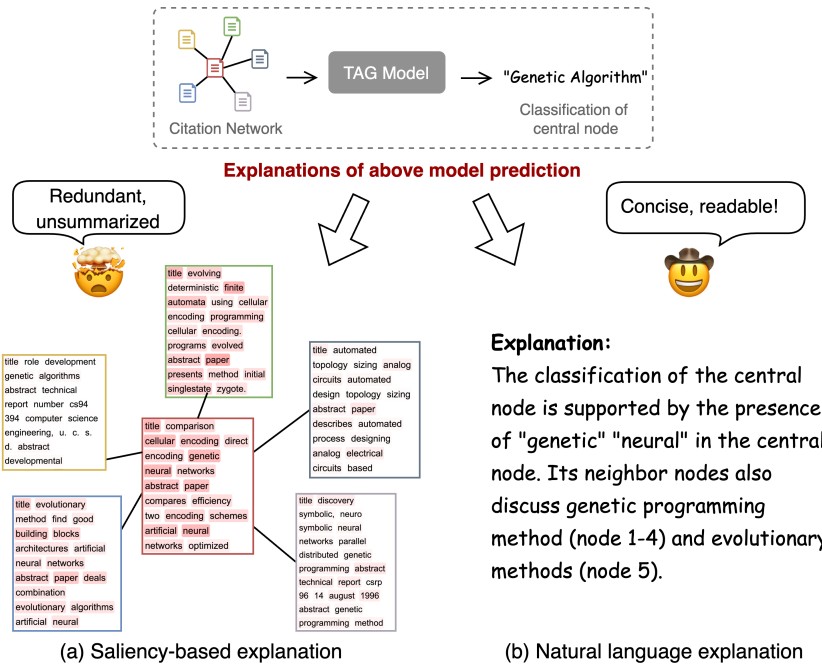

(a) Saliency-based explanation         (b) Natural language explanation

Figure 1: Illustration of saliency-based TAG explanation and natural language TAG explanation.

In this work, we present TAGExplainer, the first method to generate natural language explanations for TAG learning. Our goal is to have a generative language model that can be seen as a mapping from input-output pairs to explanations, and the generated explanations should faithfully reflect the decision-making process and be friendly for humans to understand. This goal cannot be achieved by zero-shot querying an LLM since it has no access to the model's internal decision-making process. Therefore, it is necessary to fine-tune the generative model with the labels of explanation. Since in real-world scenarios, it is impractical to have sufficient amount of annotated ground truth data for explaining model behaviors, we prompt an LLM with saliency-based explanations to generate natural language explanation pseudo-labels. To improve the quality of pseudo-labels, we propose three training objectives related to faithfulness and brevity, and iteratively fine-tune the pseudo-label generator model with these objectives with expert iteration. Finally, the generated pseudo-labels are used to train an end-to-end explainer model, which serves as our end-to-end explanation generation model.

Our contribution can be summarized as follows:

- We propose TAGExplainer, a novel framework to generate natural language explanations for TAG learning models. TAGExplainer is a model-agnostic black-box model explanation method that can generate post-hoc explanations based on the input and prediction.
- We propose to narrate saliency-based TAG explanations by our TAG explanation verbalization and prompting. Our graph verbalization preserves the semantic information, structural information as well as feature importance, and forms a more LLM-understandable format of input to be prompted the explanation pseudo-label generator LLM.

- We propose to iteratively self-train the explanation pseudo-label generator LLM via TAG Explanation Expert Iteration with three objectives for improving the graph explanation quality, considering the faithfulness to important input, faithfulness to output, and brevity.

- We conducted extensive experiments to validate the effectiveness of our proposed framework. Experiments show that TAGExplainer can self-improve for label generation and generate more faithful and brief explanations.

## 2 RELATED WORK

### 2.1 EXPLAINABILITY OF GRAPH NEURAL NETWORKS

GNNs have been widely adopted in fields such as social networks, molecular chemistry, and financial systems, yet their interpretability remains a significant challenge. Existing GNN explanation methods are generally categorized into instance-level and model-level approaches. Model-level methods seek to provide a broad understanding of GNN behavior independent of specific inputs. For example, ProtGNN (Zhang et al., 2022) introduces prototype learning, allowing GNNs to explain predictions through comparisons with learned prototypes, while XGNN (Yuan et al., 2020) uses reinforcement learning to generate synthetic graphs that reveal the structures driving GNN decisions. Both methods offer global insights into model behavior, emphasizing interpretability while maintaining performance. On the other hand, instance-level methods focus on explaining individual predictions by identifying important features that influence the model's decisions, primarily through feature importance. These methods include gradients/features-based approaches like SA (Baldassarre & Azizpour, 2019) and Guided BP (Baldassarre & Azizpour, 2019), perturbation-based methods such as GNNExplainer (Ying et al., 2019) and PGExplainer (Luo et al., 2020), as well as decomposition techniques like LRP (Baldassarre & Azizpour, 2019; Schwarzenberg et al., 2019) and surrogate models such as GraphLime (Huang et al., 2022). While these methods provide valuable insights into model behavior, they rely on graphical or numerical outputs and do not generate natural language explanations (NLE). Our proposed framework also focuses on instance-level explanations, aiming to enhance the interpretability of individual predictions while addressing current limitations in generating NLE. While significant progress has been made in improving the explainability of GNNs, current instance-level methods are limited to providing feature importance-based explanations, which is difficult for humans to understand when the input comes to TAGs. To our best knowledge, no efforts have been made to generate natural language explanations of graph models.

### 2.2 NATURAL LANGUAGE EXPLANATION

The traditional explanation methods for NLP, such as feature importance and saliency maps (Lei et al., 2016; Yu et al., 2019), often fall short in providing human-interpretable insights, motivating the development of more intuitive approaches like natural language explanations (Cambria et al., 2023). These methods provide textual justifications for model predictions, aiming to bridge the gap between model decisions and human understanding (Camburu et al., 2018; Rajani et al., 2019; Narang et al., 2020). A notable example is self-explanation models, where models like those studied by Wiegreffe et al. (2020) predict labels while simultaneously generating explanations in natural language. Further advancements, such as STaR (Zelikman et al., 2022), incorporate ground truth answers to refine explanations when model predictions are incorrect. With the advent of LLMs, Chain-of-Thought prompting (Wei et al., 2022) and zero-shot reasoning (Kojima et al., 2022) have enhanced their self-explanation abilities by generating coherent, step-by-step explanations. Additionally, LLMs' role as explainers for both their own predictions and other models' outputs has significantly expanded (Kroeger et al., 2023; Gat et al., 2023; Martens et al., 2023). However, LLM-generated explanations are criticized as not able to reflect the internal model behaviours (Agarwal et al., 2024; Parcalabescu & Frank, 2024).

## 3 PROBLEM FORMULATION

In this work, we delve into the task of explaining predictions for TAG models with natural languages. Formally, a TAG can be represented as $\mathcal{G} = (\mathcal{V}, A, \mathcal{X})$, where $\mathcal{V} = \{v_0, v_1, ..., v_{N-1}\}$ is a set of

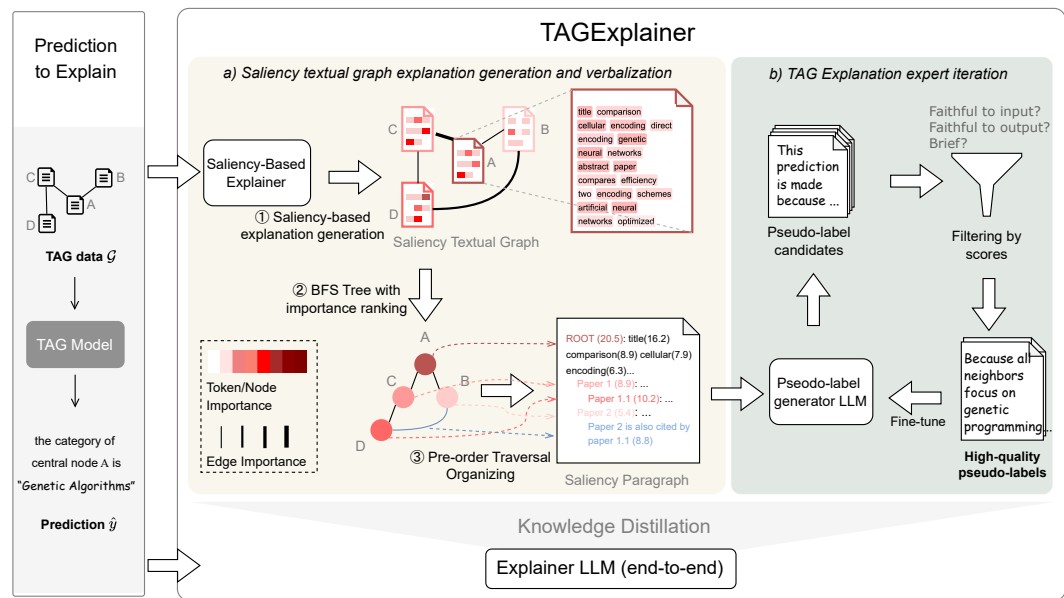

Figure 2: An illustration of TAGExplainer. A pseudo-label generator model is first trained to provide pseudo-labels, which are used for knowledge distillation to an LLM as an end-to-end explainer. (a) TAGExplainer first generates saliency-based TAG explanations, then verbalizes them into a documented form for easier understanding of LLMs, and feeds them to LLMs to generate initial natural language explanations. (b) We propose TAG explanation verbalization to convert saliency-based textual graph explanations to a documented form while preserving the semantic, structural, and saliency information. (c) We propose the TAG explanation expert iteration procedure to iteratively improve the pseudo-label generator LLM with three objectives related to faithfulness and brevity.

$N$ nodes, $A \in \{0, 1\}^{N \times N}$ is the adjacency matrix, and $\mathcal{X} = \{x_0, x_1, ..., x_{N-1}\}$ is the set of texts where $x_k = (t_{k,0}, t_{k,1}, ..., t_{k,n_k^{(t)}})$ is a sequence of tokens associated with node $v_k \in \mathcal{V}$. A TAG model $f$ is a model that can make predictions on TAGs by $\hat{y} = f(\mathcal{G})$, where $\hat{y}$ is the model output.

Given a text-attributed graph $\mathcal{G}$ and a trained TAG model $f$, the goal is to learn a mapping $g : (\mathcal{G}, \hat{y}) \to E$ to generate a paragraph of text $E$ to explain the decision-making process of $\hat{y} = f(\mathcal{G})$. The generated explanation $E$ should faithfully explain the reason for model predictions and be easy for humans to understand.

## 4 METHOD

The overall framework of TAGExplainer is illustrated in Figure 2. The high-level idea is we first train a *Pseudo-Label Generator LLM* to generate high-quality explanation labels, then the generated pseudo-labels are used to fine-tune an *Explainer LLM*. In the pseudo-labels generation phase, we first attain saliency-based TAG explanations with a saliency-based explainer, then verbalize it to the form of a *Saliency Paragraph*, and pass the *Saliency Paragraph* to the *Pseudo-Label Generator LLM*. The *Pseudo-Label Generator LLM* can be iteratively self-trained via Expert Iteration based on our proposed three training objectives to improve the quality of generated explanations. Finally, the generated high-quality pseudo-labels from the fine-tuned *Pseudo-Label Generator LLM* are used to train the end-to-end *Explainer LLM*. The details of our proposed TAGExplainer framework will be introduced in the following.

### 4.1 SALIENCY-BASED TAG EXPLANATION GENERATION AND VERBALIZATION

Due to the lack of ground truth explanation labels, we propose to generate pseudo-labels from a pre-trained LLM. Generally, we prompt this *Pseudo-Label Generator LLM* with saliency-based explanations and ask it to generate explanation candidates based on them. As illustrated in Fig. 2 (a),

specifically, a saliency-based explainer is first adopted to get the *Saliency Textual Graph* explanations, and then we transform the graph explanation into textual forms with two steps of *BFS Tree Construction with Importance Ranking* and *Pre-Order Traversal Organizing*. Finally, we prompt them to the *pseudo-label generator LLM* to get the initial explanation of pseudo-label candidates. Such a procedure will be introduced in details as follows.

**Saliency-based explanation generation.** As shown in Fig. 2 (a) Step ①, the saliency-based explanations are generated by a feature importance-based post-hoc explainer. They are represented in the form of the importance score of each node, edge, and token. An example is illustrated in Fig. 2 (a) as *Saliency Textual Graph*, where the red color from light to dark denotes the importance of nodes and tokens, and the boldness of the links between nodes denotes the edge importance. Note that here the saliency-based explainer can be any explainer that can generate post-hoc explanations for a TAG model, such as the widely used model-agnostic explanation methods including LRP, Input×Grad, Saliency, etc., so TAGExplainer is a model-agnostic framework for explaining TAG learning.

**Saliency textual graph verbalization.** Since the generated *Saliency Textual Graph* is naturally graph-structured data, it's vital to transform it into a form that LLMs are easier to understand. Therefore, we propose saliency textual graph verbalization, to transform the saliency-based graphs explanation into a document-like *Saliency Paragraph* to let LLMs better understand it, without losing any structural, semantic or feature importance information. Our verbalization methods include two steps, *1) BFS Tree Construction with Importance Ranking* and *2) Pre-Order Traversal Organizing*, as shown in steps ② and ③ in Fig. 2 (a).

*1) BFS Tree Construction with Importance Ranking*. In a TAG model prediction, the structure of a node and its k-hop salient nodes can be represented as an ego graph, with the node itself as the root. Using Breadth-First Search (BFS), this ego graph can be decomposed into a hierarchical tree, as illustrated in step ② in Figure 2 (a). Note that there can be a set of *cross-edges* that connect nodes in the BFS tree, which is illustrated as the blue link in the above figure connecting nodes B and D. During BFS, we rank the successors of the same node based on their node-level saliency scores, where nodes with higher saliency scores appear earlier (in the left side of children nodes). This is to ensure important children nodes appear earlier in the following paragraph organizing procedure and be read earlier by the *Pseudo-Label Generator LLM*.

*2) Pre-order Traversal Organizing.* We adopt a Pre-Order Traversal-based procedure to organize the tree structure into a document-like paragraph, as illustrated in step ③ in Figure 2 (a). To organize them into a document-like saliency paragraph and keep the hierarchical structure of nodes, we adopt the Pre-Order Traversal of our constructed tree structure (first visit the root, then the left subtree, then the right subtree). Each node's text is represented as a section in the document, and the successors of a node are represented as subsections. The *cross-edges* are verbalized by adding references at source nodes to the sections containing their respective destination nodes if the destination node appears earlier in the traversal, as illustrated as the blue sentence in the *Saliency Paragraph* in Figure 2 (a). This approach ensures that the document faithfully reflects the graph's structure.

When organizing the *Saliency Paragraph*, we attach the saliency information both explicitly and implicitly to prompt the *Pseudo-Label Generator LLM* with the feature importance information. Firstly, as mentioned earlier, for nodes belonging to the same parent node, we re-rank them based on their node-level importance score from higher to lower. For tokens in nodes, we attach their importance score after the tokens to prompt with the importance information, in the form of `token(score)`, without perturbing the semantic order of tokens. For all cross-edges, we also attach their importance score when mentioning them. An example of the *Saliency Paragraph* is also given in Figure 2 (a).

## 4.2 TAG EXPLANATION EXPERT ITERATION FOR EXPLANATION SELF-IMPROVEMENT

The generated *Saliency Paragraph* (introduced above) is passed to the *Pseudo-Label Generator LLM* with a prompt to explain the structure and ask it to generate explanations based on the TAG features and saliency information, which serves as our initial set of pseudo-label candidates. However, such a way cannot ensure we get high-quality pseudo-labels. Therefore, we propose to asymptomatically improve the text explanation quality by an explanation expert iteration framework that iterates the **text explanation quality measuring**, **high-quality text explanation selecting**, and **text explanation updating**, as illustrated in Fig. 2 (b). Such an iterative process is engined by our proposed

information-theoretic quality measurement that quantifies the trade-off among text explanations' faithfulness to input and prediction, as well as their brevity.

### 4.2.1 INFORMATION-THEORETIC TEXT EXPLANATION MEASUREMENTS

The generated explanation $E$ should be faithful to the model decision process, and friendly for humans to understand. To meet such requirements, we propose our TAG explanation measurements, including faithfulness to important inputs, faithfulness to outputs, and brevity.

**TAG explanation's faithfulness to important inputs.** Suppose the true rationale of the model prediction is $\mathcal{R}$, which is a subset of important nodes and the important tokens on each important node of the input $\mathcal{G}$. We want the explanation $E$ to include enough necessary information about $\mathcal{R}$. So naturally, we measure its faithfulness to $\mathcal{R}$ as the Pointwise Mutual Information (PMI) between $E$ and $\mathcal{R}$, namely,

$$f_S = \text{PMI}(E, \mathcal{R}) = \log \frac{P(\mathcal{R}, E)}{P(\mathcal{R})P(E)} = \log \frac{P(\mathcal{R}|E)}{P(\mathcal{R})} \tag{1}$$

However, estimating $P(\mathcal{R})$ and $P(\mathcal{R}|E)$ is intractable due to the big space of $\mathcal{R}$ to explore. Therefore, we propose to mask the important tokens in $\mathcal{R}$ in the original input subgraph $S$, and turn this problem into a tractable masked token recovering problem with a Masked Language Model (MLM), so $P(\mathcal{R})$ and $P(\mathcal{R}|E)$ can be estimated with $P_{MLM}(\mathcal{R}|S_M)$ and $P_{MLM}(\mathcal{R}|S_M, E)$, where $S_M$ denotes the remaining part in $S$ after masking $\mathcal{R}$ from $S$. Moreover, we also need to consider the fact that not all the tokens in $\mathcal{R}$ are equally important, so we want to prioritize the faithfulness to those most important ones, though the threshold of being important is unknown. Therefore, we sample a threshold $\tau$ in each iteration denoting the ratio of tokens in $\mathcal{S}$ to be considered as important (e.g. $\tau = 0.1$ means $\mathcal{R}$ includes the tokens with top 10% high saliency scores in $S$), then we marginalize out the influence of threshold by the following mutual information that is the probabilistic average of the above PMI as:

$$f_S = \log \frac{P(\mathcal{R}|E)}{P(\mathcal{R})} \approx \int_0^1 P(\tau) \cdot \log \frac{P_{MLM}(\mathcal{R}_\tau|S_{M_\tau}, E)}{P_{MLM}(\mathcal{R}_\tau|S_{M_\tau})} d\tau \tag{2}$$

where $P(\tau)$ is the distribution of sampling $\tau$, which can be implemented by any distribution that focus on different thresholds (e.g. the uniform distribution from 0 to 0.3), $\mathcal{R}_{.tau}$ and $S_{M_\tau}$ are the masked rationale $\mathcal{R}$ and the remaining text under the threshold $\tau$.

**TAG explanation's faithfulness to predictions.** In addition to faithfulness to important inputs, we also encourage the generated explanations to be faithful to the outputs. Similarly, for faithfulness to predictions, we leverage the PMI between explanation $E$ and the predicted label $\hat{y}$ as a measurement as

$$f_F = \text{PMI}(E, \hat{y}) = \log \frac{P(\hat{y}, E)}{P(\hat{y})P(E)} = \log \frac{P(\hat{y}|E)}{P(\hat{y})} \tag{3}$$

where $\hat{y}$ denotes the output prediction of the model to explain. The calculation of $P(\hat{y}|E)$ and $P(\hat{y})$ is also implemented with a pre-trained language model.

**TAG explanation's brevity.** Since the above objective of faithfulness encourages $E$ to be informative, which may result in generating long explanations. However, long and redundant explanations are not easy for humans to understand. Therefore, we also encourage the generated explanation $E$ to be concise, as

$$f_B = \frac{|E|}{|S|} \tag{4}$$

where $S$ is the textual form of the input TAG, which can be constructed with any graph-to-text methods such as JSON, etc. $|E|$ and $|S|$ denotes the length of $E$ and $S$.

Combining these measurements, we are essentially doing a multi-objective optimization problem, where we maximize $f_S$, $f_F$, and minimize $f_B$, which serves as our overall objective and be optimized with the following TAG expert iteration procedure of our framework.

### 4.2.2 TAG EXPLANATION EXPERT ITERATION

To effectively optimize the objectives to improve the *Pseudo-Label Generator LLM*, we propose a TAG explanation iterative training method based on Expert Iteration (Dong et al., 2023; Gulcehre

et al., 2023), as shown in Fig. 2 (b). Specifically, the training is composed of a loop of *text explanation quality measuring*, *high-quality text explanation selecting*, and *text explanation updating*, as introduced in details as follows:

(1) Text explanation quality measuring. Aligned with our training objectives, we calculate the scores $f_S$, $f_F$ and $f_B$ of the generated explanation pseudo-label $E$ based on Eq. 2, Eq. 3 and Eq. 4.

(2) High-quality text explanation selecting. Among all generated explanations, a subset of high-quality explanations is selected from all candidates with customizable criteria for adjusting the trade-off of $f_S$, $f_F$, and $f_B$, such as weighted sum and top-k.

(3) Text explanation updating. The selected high-quality explanations are used to fine-tune the *Pseudo-Label Generator LLM*. Then the model generates a new batch of explanation candidates, and it goes back to step (1).

Such three steps form a closed loop, allowing us to iteratively increase the performance of the model. Finally, we got the *Pseudo-Label Generator LLM* fine-tuned to generate faithful and brief explanations with input as the saliency paragraph.

### 4.3 END-TO-END EXPLAINER TRAINING VIA KNOWLEDGE DISTILLATION

The above loop gives us a model expert in generating high-quality explanations based on saliency-based explanations. However, our goal is to be an end-to-end explainer model that can generate natural language explanations based on the raw input and its prediction. Therefore, after fine-tuning, we distill the whole pipeline to the *Explainer LLM*. The distillation is conducted by generating a set of candidates with the pseudo-label generation pipeline, filtering the high-quality ones and using the filtered dataset to fine-tune the *Explainer LLM*.

## 5 EXPERIMENTS

### 5.1 EXPERIMENTAL SETUP

**Datasets.** We use three real-world TAG datasets, including two citation networks (Cora (Yang et al., 2016) and DBLP (Tang et al., 2008)), and one E-commerce co-purchasing network (Book-History (Yan et al., 2023)), to evaluate the performance of our method. Cora is a widely used citation network where each node represents a research paper, and edges represent citation links between papers, with each paper assigned to one of several topics. DBLP dataset is a comprehensive bibliography in computer science containing metadata on academic papers, authors, and collaborations. We extracted the top 30 most frequently occurring categories from DBLP, along with their corresponding papers. The Book-History dataset is an E-commerce co-purchasing network where nodes represent books, and edges indicate books frequently purchased or browsed together, with each book classified into different categories. More details of the datasets are in Appendix B.

**Compared Methods.** To our best knowledge, no existing method can generate natural language explanations for graph learning. The most relevant method is SMV (Feldhus et al., 2022), which is designed to verbalize saliency map explanations for text classification models. To evaluate the effectiveness of TAGExplainer, we compare it with various most advanced LLMs to generate explanations in a zero-shot manner given the input subgraph and the model prediction. We benchmarked our method with the most advanced LLMs, including GPT-4o, GPT-3.5 turbo, LLaMA 3, and SMV method based on GPT-4o (denoted as SMV in the results).

**Evaluation Metrics.** For an overall evaluation of the quality, we evaluate the faithfulness and brevity of explanations. Following previous research (Padmakumar & He, 2021; Li et al., 2020), *Pointwise Mutual Information* (PMI) and *Simulatability* (Simul.) are used as indicators for faithfulness. PMI (Padmakumar & He, 2021; Chen et al., 2022; Colombo et al., 2022; Darrin et al., 2024) measures the mutual information between the generated explanations and the important regions in the input text. The top 10%, 20%, and 30% important tokens are used as references for the calculation of PMI, denoted as PMI-10%, PMI-20%, and PMI-30% in the result tables. *Simulatability* (Sushil et al., 2018; Sia et al., 2023; Li et al., 2020; Pruthi et al., 2022) measures the accuracy of the model prediction

| Dataset | Method | Metrics | | | | |
|---|---|---|---|---|---|---|
| | | Simul. (↑) | PMI-10% (↑) | PMI-20% (↑) | PMI-30% (↑) | Brevity (↓) |
| Cora | LLaMA3.1 8B | 0.78 | 0.335 | 0.278 | 0.199 | 0.600 |
| | GPT-3.5 Turbo | 0.83 | 0.340 | 0.281 | 0.213 | 0.318 |
| | GPT-4o | 0.95 | 0.414 | 0.284 | 0.225 | 0.357 |
| | SMV | 0.88 | 0.359 | 0.267 | 0.217 | 0.431 |
| | TAGExplainer | **0.97** | **0.418** | **0.290** | **0.227** | **0.315** |
| DBLP | LLaMA3.1 8B | 0.63 | 0.139 | 0.109 | 0.077 | 0.394 |
| | GPT-3.5 Turbo | 0.71 | 0.136 | **0.110** | 0.084 | 0.403 |
| | GPT-4o | 0.82 | 0.142 | 0.101 | 0.085 | 0.385 |
| | SMV | 0.76 | 0.139 | 0.098 | 0.082 | 0.419 |
| | TAGExplainer | **0.95** | **0.155** | 0.108 | **0.085** | **0.354** |
| Book-History | LLaMA3.1 8B | 0.79 | 0.465 | **0.390** | 0.281 | 0.735 |
| | GPT-3.5 Turbo | 0.83 | 0.436 | 0.361 | 0.270 | 0.853 |
| | GPT-4o | 0.89 | 0.456 | 0.313 | 0.240 | 0.768 |
| | SMV | 0.87 | 0.441 | 0.320 | 0.257 | 0.836 |
| | TAGExplainer | **0.96** | **0.533** | 0.374 | **0.291** | **0.506** |

Table 1: The main results of quality of natural language explanations generated by different methods. Best results are bolded.

can be correctly inferred from the explanation. For *Brevity*, the average ratio of explanation length and input length is used as an indicator.

**Implementation Details.** We used the GPT-4o-mini-2024-07-18 model with our prompt for TAG explanation for generating candidate explanations (the details of prompts are given in Appendix E), applying one-shot learning for consistency. For scoring and rejection sampling, we utilized the fine-tuned gemma2-2b-it model to estimate the conditional probability distribution. We applied a balanced configuration for the three objectives (we select pseudo-label candidates whose three scores are all among the top 50% of all candidates generated in that iteration). In each iteration, 50 high-quality samples were selected for fine-tuning via the OpenAI API with default settings. Finally, we used the fine-tuned LLaMA-3.1-8b as the base student model for knowledge distillation using the LoRA technique. More implementation details are given in Appendix D.

## 5.2 QUANTITATIVE EVALUATION

**Quality of generated explanations (main results).** Our experimental results, shown in Table 1, demonstrate that TAGExplainer consistently performs well in generating high-quality explanations. Specifically, TAGExplainer shows an 8.2% average improvement over the second-best performer in the PMI-10% metric over three datasets, highlighting its effectiveness at capturing the most important information for model decisions. In the few cases where it does not rank first, TAGExplainer's performance remains very close to the best, illustrating its robustness. In terms of simulatability, TAGExplainer outperforms all baseline methods by 8.6%, achieving a simulatability score of 0.95 across all three datasets, significantly higher than other methods, proving highly faithful to model predictions. For the brevity metric, TAGExplainer is 13.4% better than the second-best performer, effectively balancing conciseness and accuracy. Across all three datasets, TAGExplainer generates relatively compact explanations while maintaining high simulatability scores. These results demonstrate that TAGExplainer successfully navigates the inherent trade-offs among PMI, simulatability, and brevity. It consistently produces explanations that align closely with model predictions while remaining concise, further enhancing their interpretability. This balance highlights TAGExplainer's strength in delivering both faithful and interpretable explanations. Among the baseline methods, we observed that larger models, like GPT-4o, tend to achieve higher simulatability and brevity scores compared to smaller models, while all LLMs perform similarly in the PMI metric due to their limited access to the model's internal decision-making process, which constrains their ability to fully explain model predictions. TAGExplainer overcomes this limitation by directly referencing the decision process, giving it a distinct advantage in generating more faithful and interpretable explanations.

**Effectiveness of TAG Explanation Expert Iteration.** The training score curve in our iterative training (Expert Iteration) of the pseudo-label generator LLM is illustrated in Figure 3. With an increasing number of iterations, both faithfulness to important inputs and faithfulness to predictions

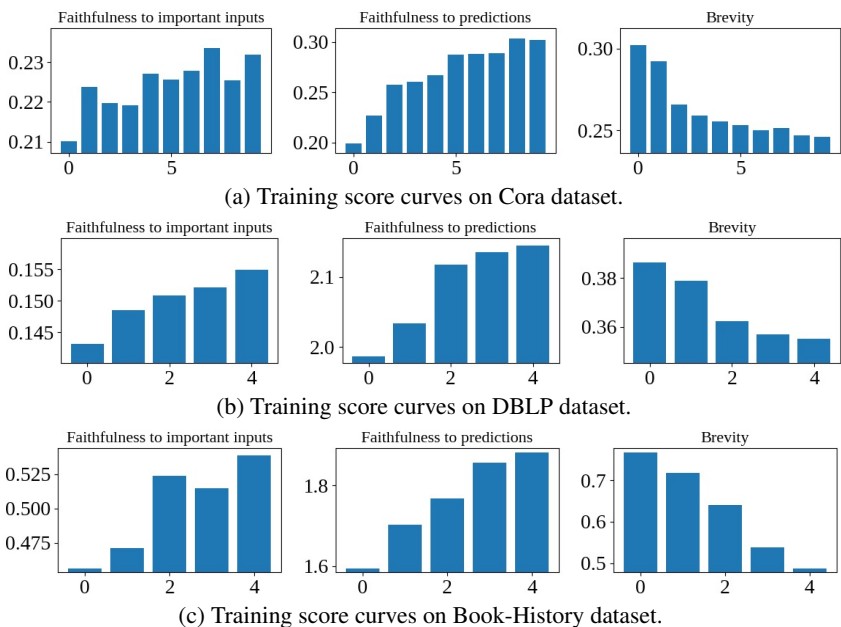

(a) Training score curves on Cora dataset.

(b) Training score curves on DBLP dataset.

(c) Training score curves on Book-History dataset.

Figure 3: The change of three pseudo-label quality scores in the TAG explanation expert iteration process, w.r.t to each training iteration. The X-axis represents the number of iterations, while the Y-axis represents the value of each corresponding score (shown above each plot).

| Method | Metrics | | | | |
|---|---|---|---|---|---|
| | PMI-10% (↑) | PMI-20% (↑) | PMI-30% (↑) | Simul. (↑) | Brevity (↓) |
| TAGExplainer | 0.418 | 0.290 | 0.227 | 0.97 | 0.315 |
| w/o $f_S$ | 0.407 | 0.298 | 0.213 | 0.98 | 0.304 |
| w/o $f_F$ | 0.419 | 0.311 | 0.241 | 0.90 | 0.315 |
| w/o $f_B$ | 0.432 | 0.327 | 0.239 | 0.96 | 0.361 |
| w/o Expert Iteration | 0.414 | 0.284 | 0.225 | 0.95 | 0.357 |

Table 2: Results of ablation study. TAGExplainer denotes our method, w/o $f_S$, $f_F$, $F_B$ denotes removing the corresponding objective for training. w/o Expert Iteration means removing the expert-iteration of pseudo-label generator LLM. The cells highlighted in red in the table represent metrics where the performance is expected to drop when the corresponding component is removed.

exhibit a overall upward trend, while brevity shows a gradual decline. This iterative learning process underscores our TAG explanation expert iteration's effectiveness in gradually and iteratively improving the faithfulness to important input, faithfulness to output, and brevity. It is worth highlighting that during each iteration of the expert iteration process, we select only 50 high-quality samples. Despite this small sample number, our TAG explanation expert iteration consistently enhances faithfulness to both important inputs and predictions. This demonstrates that the process is highly efficient, as steady improvements in performance are achieved by using just a small, carefully chosen set of high-quality samples in each iteration. Additional quantitative analyses are provided in Appendix A, including the study of different pseudo-label selection strategies and the performance improvement of the explainer LLM after fine-tuning with generated high-quality pseudo-labels.

## 5.3 QUALITATIVE EVALUATION

As depicted in Figure 4a, we visualize the token importance of saliency-based explanation, where individual words within each node are highlighted with varying intensities of red to indicate their saliency scores. Darker red hues correspond to higher saliency scores, while lighter shades represent lower ones. In contrast, Figure 4b showcases the natural language explanations generated by TAG-Explainer, with key terms such as "reinforcement learning" and "learning algorithm" emphasized in yellow. This demonstrates TAGExplainer's ability to not only capture the salient information identi-

ROOT: title emergent hierarchical control structures learning reactive hierarchical relationships reinforcement environments abstract

Node-1: title transfer learning composing solutions elemental sequential tasks abstract although building sophisticated learning agents

Node-1.1: title modular q learning architecture manipulator task decomposition data storage cerebellar model ar abstract

Node-1.2: title efficient learning multiple task sequences abstract present modular network architecture learning algorithm based incremental

Node-1.3: title robust reinforcement learning motion planning abstract exploring find better solutions, agent performing online reinforcement

Node-1.4: title learning achieve goals abstract temporal difference methods solve temporal credit assignment problem reinforcement learning.

Node-1.5: title coevolutionary approach learning sequential decision rules abstract present coevolutionary approach learning sequential decision rules

Node-1.6: title hierarchical reinforcement learning maxq value function decomposition abstract paper describes maxq method hierarchical

Node-1.7: title using case based reasoning reinforcement learning framework optimization changing criteria abstract practical optimization problems job

Node-1.8: title hierarchical explanation based reinforcement learning abstract explanation based reinforcement learning ebrl introduced dietterich

Node-2: title learning continuous domains delayed rewards abstract much done develop learning techniques delayed reward problems

Node-2.1: title generalization reinforcement learning successful examples using sparse coarse coding abstract large problems, reinforcement learning

Node-3: title learning hierarchical control structures multiple tasks changing environments abstract need hierarchies tasks changing environments

(a) Saliency-Based Explanation

The classification of the ROOT node into the "Reinforcement Learning" category can be explained as follows:

ROOT mentions reinforcement learning, learning algorithms, and hierarchical control structures indicating that it focuses on reinforcement learning methodologies and their applications.
- Node-1 discusses transfer learning in reinforcement learning, highlighting how learning agents can adapt to multiple tasks.
 - Node-1.1 ~ Node-1.8 provide more context about various reinforcement learning architectures and approaches, including modular Q-learning and hierarchical reinforcement learning.
- Node-2 focuses on learning in continuous domains with delayed rewards, a pertinent aspect of reinforcement learning, particularly in real-world applications.
 - Node-2.1 discusses generalization in reinforcement learning using neural networks, which is relevant but less directly tied to the prediction.
- Node-3: This node discusses the need for hierarchies within control systems and the improvements in nested Q-learning, which are highly relevant to the classification label.

In summary, the classification of the ROOT node into "Reinforcement Learning" is well-supported by the strong presence of key reinforcement learning terms within the node itself and its direct connections to Nodes 1, 2 and 3, which also focus on reinforcement learning methodologies.

(b) natural language Explanation

Figure 4: Visualization of a saliency-based explanation and a corresponding natural language explanation generated by TAGExplainer. In (a), red words indicate important terms, with darker red showing higher importance. In (b), yellow highlights reference high-saliency areas and emphasize that the explanation summarized key information.

fied in the saliency map but also present it in a more accessible and interpretable manner. Compared with the saliency-based explanations that merely highlight important words, TAGExplainer goes beyond synthesizing and abstracting content across nodes. For example, in the case of Node-1.1 through Node-1.8, TAGExplainer effectively integrates the most relevant information into a coherent explanation rather than simply reproducing the input. This showcases TAGExplainer's strength in generating explanations that are more informative and contextualized than the visual saliency approach. Additional examples are provided in Appendix C.

## 5.4 Ablation Study

In the ablation study, we removed $f_S$, $f_F$, $f_B$ and Expert Iteration to test their corresponding effectiveness, as shown in Table 2. The results highlight the effectiveness of different components of the proposed framework. Removing the saliency optimization objective (w/o $f_S$) leads to a decrease in PMI scores, proving its importance for relevance. Excluding the fidelity objective (w/o $f_F$) results in lower simulatability, and removing the brevity optimization objective (w/o $f_B$) results in decreases brevity, proving its role in keeping explanations concise. Removing the self-training process (w/o Expert Iteration) leads to declines in all metrics, underscoring its importance for iterative refinement of the proposed framework. Notably, when one objective is removed, the other two often improve. For instance, removing the saliency objective (w/o $f_S$) decreases PMI scores, but simulatability improves and brevity shortens. This is because the three objectives inherently involve trade-offs—removing one allows the remaining two to be optimized within a larger space, free from the constraints imposed by the removed objective, naturally leading to better performance in those areas.

## 6 Conclusion

In this paper, we present TAGExplainer, a model-agnostic post-hoc explainer to generate natural language explanations for TAG learning models. TAGExplainer fine-tunes a generative language model as an explanation generator with pseudo-labels derived from saliency-based explanations. Through iterative self-training, we improve the quality of generated explanation pseudo-labels, ensuring the explanation generator can be trained with high-quality data. Our extensive experiments demonstrate the effectiveness of TAGExplainer.

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

# A  ADDITIONAL EXPERIMENT RESULTS

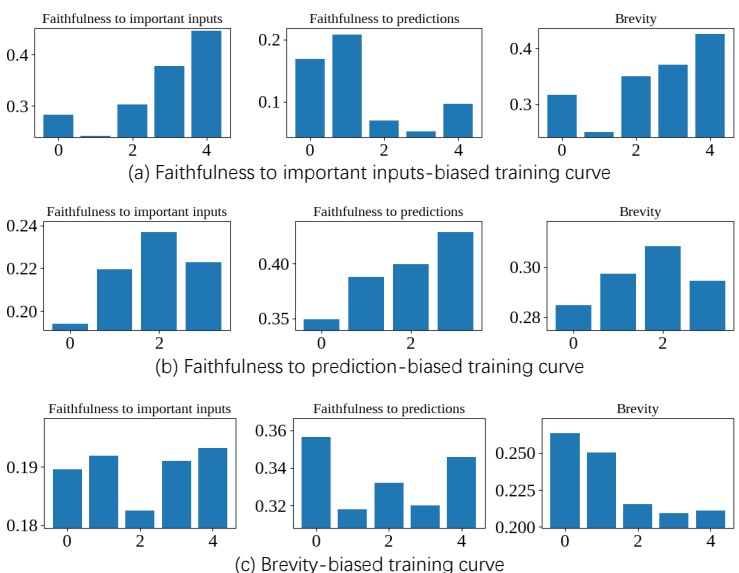

Figure 5: experiment with different customized selecting criteria

## A.1  EXPERIMENTS ON DIFFERENT CANDIDATE SELECTION CRITERIA

To provide an initial validation of the effectiveness of the optimization process for the explanation generator, we conducted experiments using three extreme selection criteria: (a) a selection strategy that prioritizes only faithfulness to important inputs, (b) a strategy focusing exclusively on faithfulness to predictions, and (c) a strategy considering solely brevity. Our results (as shown in Figure 5) indicate that, under each of these conditions, the corresponding metric was significantly improved. These findings suggest that the proposed framework has the capacity to selectively enhance the performance of the explanation generator with respect to specific evaluation metrics, demonstrating its adaptability and targeted optimization potential.

## A.2  THE PERFORMANCE GAIN OF EXPLAINER LLM AFTER KNOWLEDGE DISTILLATION

| Dataset | Method | Metrics | | | | |
|---|---|---|---|---|---|---|
| | | PMI-10% (↑) | PMI-20% (↑) | PMI-30% (↑) | Simul. (↑) | Brevity (↓) |
| Cora | LLaMA3.1 8B | 0.335 | 0.278 | 0.199 | 0.78 | 0.600 |
| | TAGExplainer | **0.418** | **0.290** | **0.227** | **0.97** | **0.315** |
| DBLP | LLaMA3.1 8B | 0.139 | **0.109** | 0.077 | 0.63 | 0.394 |
| | TAGExplainer | **0.155** | 0.108 | **0.085** | **0.95** | **0.354** |
| Book-History | LLaMA3.1 8B | 0.465 | **0.390** | 0.281 | 0.79 | 0.735 |
| | TAGExplainer | **0.533** | 0.374 | **0.291** | **0.96** | **0.506** |

Table 3: The performance of student model before (LLaMA3.1 8B) and after (our TAGExplainer) distillation. Better results are bolded.

We tested the performance of vanilla LLaMA3.1 8B model and the distilled version, which leads to our TAGExplainer, among three different datasets. The results demonstrated in table 3 shows that the distillation process indeed promoted the quality of explanation in terms of most PMI, Simulatability and Brevity metrics.

# B   DATASET DETAILS

|  | # Nodes | # Edges | # Categories |
|---|---|---|---|
| Cora | 2,708 | 5,429 | 7 |
| DBLP | 110,757 | 655,766 | 30 |
| Book-History | 41,551 | 358,574 | 12 |

Table 4: Dataset Overview

We conduct experiments on 3 datasets, the basic statistics are shown in Table 4.

Cora is a network that contains computer science research papers, where each node represents a paper, and each edge represents one paper and cites the other one. Nodes in the Cora dataset are classified into seven categories: Case_Based, Genetic_Algorithms, Neural_Networks, Probabilistic_Methods, Reinforcement_Learning, Rule_Learning, and Theory.

DBLP dataset is a large-scale network of academic research papers, where each node represents a paper and each edge indicates a citation between two papers. Similar to the Cora dataset, which focuses on computer science research, DBLP covers a broader range of fields of study with an emphasis on computer science and related disciplines. Papers in the DBLP dataset are classified into various fields of study based on their topics. From the DBLP dataset, we extracted the top 30 most frequently occurring fields of study, along with their corresponding papers. Some of these categories include cluster analysis, cloud computing, computer science, the internet, wireless sensor networks, artificial neural networks, population, control theory, image segmentation, humanities, and image processing. These categories reflect the diverse range of research areas covered in the DBLP dataset.

Book-History dataset, extracted from the Amazon dataset (Ni et al., 2019), comprises items labeled under the second-level category "History." In this dataset, each node represents a book, and edges between nodes indicate frequent co-purchases or co-views of the books. The books in the Book-History dataset are classified into 12 distinct categories: Africa, Americas, Ancient Civilizations, Arctic & Antarctica, Asia, Australia & Oceania, Europe, Historical Study & Educational Resources, Middle East, Military, Russia, and World.

# C   CASE STUDY

In this subsection, we provide two examples of explanations generated by TAGExplainer. Token importance of the input is visualized using background colors, where more important tokens are shown in a deeper shade of red, while less important tokens are displayed in a lighter shade of red. For each example, we present the input with color-coded token importance, along with TAGExplainer's corresponding explanation. Examples extracted from the DBLP and Book-History datasets are shown in Fig. 6 and Fig. 7, respectively.

# D   IMPLEMENTATION DETAILS

We first masked the last $5\%$ of tokens (mostly stop words and punctuations without explicit semantical contribution to the downstream tasks) based on their importance scores to form a reduced saliency-based explanation as input. We then utilized the candidate explanation generator GPT-4o-mini-2024-07-18, prompting it with a carefully designed template (see Appendix E for details) and employing a one-shot learning technique to ensure consistency in the format and style of the generated explanations. For scoring and rejection sampling, we used the fine-tuned gemma2-2b-it model as the masked language model (MLM) to estimate the conditional probability distribution mentioned in information-theoretic objectives. In Equation 3, we have masked all label-related information in condition E to prevent answer leakage. During the rejection sampling phase, we found that a balanced configuration among all three objectives introduced in 4.2.1, i.e., $\lambda_S : \lambda_F : \lambda_B = 1 : 1 : 1$, provided stable and balanced performance across the three evaluation metrics (more customized criteria are included Appendix A). We applied self-supervised fine-tuning by selecting 50 high-quality

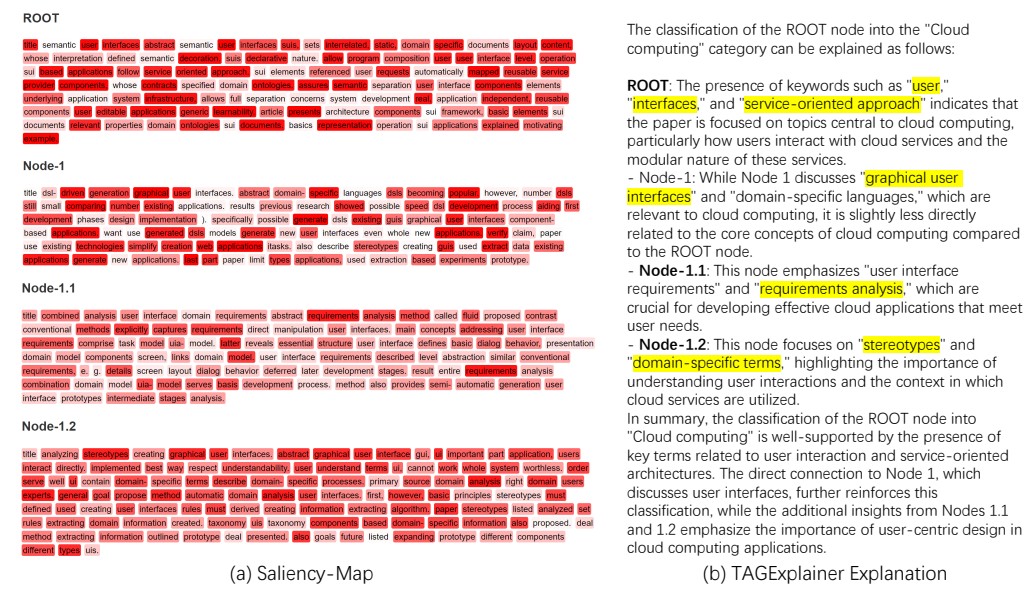

(a) Saliency-Map        (b) TAGExplainer Explanation

Figure 6: Example from DBLP

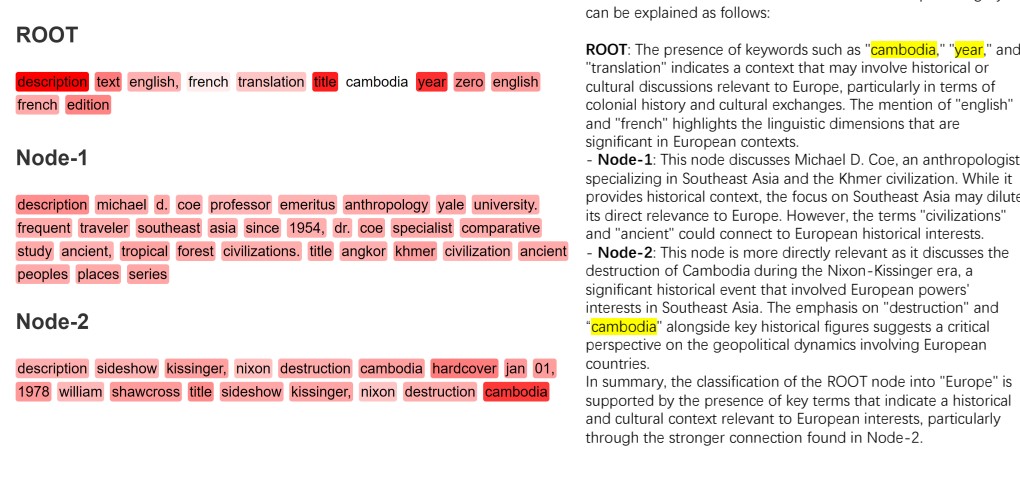

(a) Saliency-Map        (b) TAGExplainer Explanation

Figure 7: Example from Book-History

samples during each loop via rejection sampling. These samples were used to fine-tune the model using OpenAI API with default learning rate and batch size for 3 epochs. The final model obtained from the optimization loop served as the teacher model. We then performed knowledge distillation using the fine-tuned LLaMA-3.1-8b as the base student model, employing the LoRA technique (rank r=16 and alpha=16) for efficient fine-tuning. We minimized the cross-entropy loss between the student outputs and the teacher outputs, which resulted in our final TAGExplainer model. The comparative analysis of the student model's performance before and after distillation is presented in Appendix A.2

# E   PROMPT DETAILS

In our experiments, we utilize two types of prompts: one in which each token in the input is accompanied by a corresponding saliency score, and another without saliency scores. The difference between the two prompts is whether the words in the verbalized graph are accompanied by their corresponding importance scores in brackets or not.The teacher models require the inclusion of saliency scores, as they function as candidate explanation generators. The presence of saliency scores enables them to generate more accurate explanations by highlighting important tokens. In contrast, the student models do not use saliency scores; their task is to output the reasoning process of the black-box model based solely on the TAG and prediction. The student models are designed to align directly with the teacher models' outputs, ensuring consistency without requiring saliency information. All the prompts we used are given from next page, due to the limited space.

**Zero-Shot w/o Saliency Prompt Part.1**

**HumanMessage:** "The following verbalized graph contains important words in the text of each node. These words contribute to the classification of Node 0 into one of the seven possible categories (['Case Based', 'Genetic Algorithms', 'Neural Networks', 'Probabilistic Methods', 'Reinforcement Learning', 'Rule Learning', 'Theory']).

Generate a concise, human-readable explanation that justifies the classification result of Node 0 by identifying and explaining the relevant inner-node features (i.e., keywords) and inter-node relationships (i.e., graph structure). The explanation should focus on how these factors contribute to the classification label.

## Example

### Verbalized Graph

`<verbalized-graph>`

ROOT: title experiments real time decision algorithms abstract real time decision algorithms class incremental resource bounded horvitz, 89 anytime dean, 93 algorithms evaluating influence diagrams. present test domain real time decision algorithms, results experiments several real time decision algorithms domain. results demonstrate high performance two algorithms, decision evaluation variant incremental probabilisitic inference dambrosio, 93 variant algorithm suggested goldszmidt, goldszmidt, 95 ], pk reduced. discuss implications experimental results explore broader applicability algorithms.

Node-1: title learning policies partially observable environments scaling abstract partially observable markov decision processes pomdp model decision problems agent tries maximize reward face limited noisy sensor feedback. study pomdp motivated need address realistic problems, existing techniques finding optimal behavior appear scale well unable find satisfactory policies problems dozen states. brief review pomdp s, paper discusses several simple solution methods shows capable finding near optimal policies selection extremely small pomdp taken learning literature. contrast, show none able solve slightly larger noisier problem based robot navigation. find combination two novel approaches performs well problems suggest methods scaling even larger complicated domains.

Node-1.1: title formal framework speedup learning problems solutions abstract speedup learning seeks improve computational efficiency problem solving experience. paper, develop formal framework learning efficient problem solving random problems solutions. apply framework two different representations learned knowledge, namely control rules macro operators, prove theorems identify sufficient conditions learning representation. proofs constructive accompanied learning algorithms. framework captures empirical explanation based speedup learning unified fashion. illustrate framework implementations two domains symbolic integration eight puzzle. work integrates many strands experimental theoretical work machine learning, including empirical learning control rules, macro operator learning.

Node-1.2: title acting uncertainty discrete bayesian models mobile robot navigation abstract discrete bayesian models used model uncertainty mobile robot navigation, question actions chosen remains largely unexplored. paper presents optimal solution problem, formulated partially observable markov decision process. since solving optimal control policy intractable, general, goes explore variety heuristic control strategies. control strategies compared experimentally, simulation runs robot.

Node-1.3: title incremental methods computing bounds partially observable markov decision processes abstract partially observable markov decision processes pomdps allow one model complex dynamic decision control problems include action outcome uncertainty imperfect observabil ity. control problem formulated dynamic optimization problem value function combining costs rewards multiple steps. paper propose, analyse test various incremental methods computing bounds value function control problems infinite discounted horizon criteria. methods described tested include novel incremental versions grid based linear interpolation method simple lower bound method sondik updates. work arbitrary points belief space enhanced various heuristic point selection strategies. also introduced new method computing initial upper bound fast informed bound method. method able improve significantly standard commonly used upper bound computed mdp based method. quality resulting bounds tested maze navigation problem 20 states, 6 actions 8 observations.

Node-1.4: title learning sorting decision trees pomdps abstract pomdps general models sequential decisions actions observations probabilistic. many problems interest formulated pomdps, yet use pomdps limited lack effective algorithms. recently started change number problems robot navigation planning beginning formulated solved pomdps. advantage pomdp approach clean semantics ability produce principled solutions integrate physical information gathering actions. paper pursue approach context two learning tasks learning sort vector numbers learning decision trees data. problems formulated pomdps solved general pomdp algorithm. main lessons results 1 use suitable heuristics representations allows solution sorting classification pomdps non trivial sizes, 2 quality resulting solutions competitive best algorithms, 3 problematic aspects decision tree learning test mis classification costs, noisy tests, missing values naturally accommodated."

---

**Zero-Shot w/o Saliency Prompt Part.2**

**HumanMessage:** "Node-1.5: title approximating optimal policies partially observable stochastic domains abstract problem making optimal decisions uncertain conditions central artificial intelligence. state world known times, world modeled markov decision process mdp ). mdps studied extensively many methods known determining optimal courses action, policies. realistic case state information partially observable, partially observable markov decision processes pomdps ), received much less attention. best exact algorithms problems inefficient space time. introduce smooth partially observable value approximation spova ), new approximation method quickly yield good approximations improve time. method combined reinforcement learning methods, combination effective test cases.

Node-1.6: title efficient dynamic programming updates partially observable markov decision processes abstract examine problem performing exact dynamic programming updates partially observable markov decision processes pomdps computational complexity viewpoint. dynamic programming updates crucial operation wide range pomdp solution methods find intractable perform updates piecewise linear convex value functions general pomdps. offer new algorithm, called witness algorithm, compute updated value functions efficiently restricted class pomdps number linear facets great. compare witness algorithm existing algorithms analytically empirically find fastest algorithm wide range pomdp sizes.

Node-2: title efficient inference bayes networks combinatorial optimization problem abstract number exact algorithms developed perform probabilistic inference bayesian belief networks recent years. techniques used algorithms closely related network structures easy understand implement. paper, consider problem combinatorial optimization point view state efficient probabilistic inference belief network problem finding optimal factoring given set probability distributions. viewpoint, previously developed algorithms seen alternate factoring strategies. paper, define combinatorial optimization problem, optimal factoring problem, discuss application problem belief networks. show optimal factoring provides insight key elements efficient probabilistic inference, demonstrate simple, easily implemented algorithms excellent performance.

Node-2.1: title sensitivities alternative conditional probabilities bayesian belief networks abstract show alternative way representing bayesian belief network sensitivities probability distributions. representation equivalent traditional representation conditional probabilities, makes dependencies nodes apparent intuitively easy understand. also propose qr matrix representation sensitivities conditional probabilities efficient, memory requirements computational speed, traditional representation computer based implementations probabilistic inference. use sensitivities show certain class binary networks, computation time approximate probabilistic inference positive upper bound error result independent size network. finally, alternative traditional algorithms use conditional probabilities, describe exact algorithm probabilistic inference uses qr representation sensitivities updates probability distributions nodes network according messages neighbors.

Node-2.2: title algebraic techniques efficient inference bayesian networks abstract number exact algorithms developed perform probabilistic inference bayesian belief networks recent years. algorithms use graph theoretic techniques analyze exploit network topology. paper, examine problem efficient probabilistic inference belief network combinatorial optimization problem, finding optimal factoring given algebraic expression set probability distributions. define combinatorial optimization problem, optimal factoring problem, discuss application problem belief networks. show optimal factoring provides insight key elements efficient probabilistic inference, present simple, easily implemented algorithms excellent performance. also show use algebraic perspective permits significant extension belief net representation.

Node-2.3: title interpretation complex scenes using bayesian networks abstract object recognition systems, interactions objects scene ignored best interpretation considered set hypothesized objects matches greatest number image features. show image interpretation cast problem finding probable explanation mpe bayesian network models visual physical object interactions. problem determine exact conditional probabilities network shown unimportant, since goal find probable configuration objects, calculate absolute probabilities. furthermore show evaluating configurations feature counting equivalent calculating joint probability configuration using restricted bayesian network, derive assumptions probabilities necessary make bayesian formulation reasonable.

Node-2.4: title case based probability factoring bayesian belief networks abstract bayesian network inference formulated combinatorial optimization problem, concerning computation optimal factoring distribution represented net. since determination optimal factoring computationally hard problem, heuristic greedy strategies able find approximations optimal factoring usually adopted. present paper investigate alternative approach based combination genetic algorithms ga case based reasoning cbr ). show use genetic algorithms improve quality computed factoring case static strategy used mpe computation ), combination ga cbr still provide advantages case dynamic strategies. preliminary results different kinds nets reported.
`</verbalized-graph>`"

---

**Zero-Shot w/o Saliency Prompt Part.3**

**HumanMessage:** " ### Classification Label Probabilistic Methods
### Reasoning
0. Graph Structure Reconstruction:
In the provided verbalized graph, The ROOT node (first line) is the target for classification.
Single-digit indexed nodes are direct neighbors of ROOT.
Double-digit indexed nodes are:
- Two hops away from ROOT
- Direct children of their parent node
More digits indexed nodes follow the same principle as described above.

Thus, the graph structure of this verbalized graph is:

- ROOT
  - Node-1
    * Node-1.1
    * Node-1.2
    * Node-1.3
    * Node-1.4
    * Node-1.5
    * Node-1.6
  - Node-2
    * Node-2.1
    * Node-2.2
    * Node-2.3
    * Node-2.4

1. Word-Level Evaluation:
Detect important terms for the classification label.
Quantitatively, the importance (saliency) scores behind each word in the verbalized graph are calculated by a post-hoc XAI (explainable AI) algorithm.
Semantically, keywords synonymous with or closely related to label, 'Probabilistic Methods' (such as 'probabilistic', 'probability', and 'inference'), are semantically influential in determining the classification because they directly indicate the node's relevance to the 'Probabilistic Methods' category.
2. Graph-Level Aggregation:
Node 1 does not contain as many directly relevant keywords. Yet, it still discusses POMDPs, which are probabilistic models, indicating that it contributes to the classification in a less significant way.
Direct neighbors like Node 2, which contains the keyword 'probabilistic', have a substantial influence on ROOT node classification due to their close proximity to the ROOT node.
Indirect neighbors, Nodes-2.1 to Node-2.4, connected through Node 2, contain keywords like 'probabilistic' and 'probability', which further reinforce the classification of ROOT.
Meanwhile, the indirect neighbors connected through Node 1, Node-1.1 to Node-1.6, are less relevant because they do not include important keywords directly related to 'Probabilistic Methods'.

### Free-Text Explanation "'markdown The classification of ROOT node into the "Probabilistic Methods" category can be explained as follows:
ROOT: The presence of keywords like "probabilistic" and "inference" within ROOT node suggests that the paper is focused on topics central to probabilistic methods, such as performing probabilistic inference and evaluating influence diagrams.
- Node-1: While Node 1 focuses on POMDPs, which are also probabilistic models, it is slightly less directly related to the probabilistic inference methods emphasized in ROOT node.
- Node-1.1  Node-1.6: they are less relevant to probabilistic methods and thus contribute less to the classification.
- Node-2: This node discusses "probabilistic inference" and "Bayesian networks," both of which are foundational to probabilistic methods. The strong thematic connection between ROOT node and Node 2 reinforces the classification.
- Node-2.1  Node-2.4: they also discuss probabilistic topics, further supporting the classification.

In summary, the classification of ROOT node into "Probabilistic Methods" is well-supported by the strong presence of key probabilistic terms within the node itself and its direct connection to Node 2, which also focuses on probabilistic inference. Indirect neighbors through Node 2 further reinforce this classification."

**Zero-Shot w/o Saliency Prompt Part.4**

**HumanMessage:** "## Task
### Verbalized Graph
`<verbalized-graph>`{document}`</verbalized-graph>`
### Classification Label
`{label}`
### Reasoning
### Free-Text Explanation
*(P.S.: 1. make sure to complete both the reasoning section and then Free-Text Explanation section with the same structure as exemplified above.*
*2. make good use of the importance (saliency) score behind each word as your guidance to generate the better explanation. However, it is not necessary to directly quote the saliency score.*
*3. use the **whole** graph structure you constructed during reasoning for the format of the explanation. Indents and node indexes are necessary, which represent the hierarchy of the graph.)* "

**Zero-Shot w/ Saliency Prompt Part.1**

**HumanMessage:** "The following verbalized graph contains important words in the text of each nodes. These words (each with corresponding importance score in the bracket) contributes to the classification of Node 0 into one of the seven possible categories (['Case Based', 'Genetic Algorithms', 'Neural Networks', 'Probabilistic Methods', 'Reinforcement Learning', 'Rule Learning', 'Theory']).
Generate a concise, human-readable explanation that justifies the classification result of Node 0 by identifying and explaining the relevant inner-node features (i.e., keywords) and inter-node relationships (i.e., graph structure). The explanation should focus on how these factors contribute to the classification label.
## Example
### Verbalized Graph
`<verbalized-graph>`
ROOT: title(9.13) experiments(7.56) real(2.52) time(2.41) decision(5.20) algorithms(7.18) abstract(12.01) real(3.17) time(2.82) decision(5.46) algorithms(10.39) class(4.34) incremental(2.60) resource(4.50) bounded(5.79) horvitz,(2.67) 89(4.58) anytime(6.66) dean,(4.92) 93(5.03) algorithms(7.94) evaluating(4.75) influence(7.70) diagrams.(10.34) present(16.50) test(6.61) domain(10.50) real(3.11) time(2.89) decision(5.51) algorithms,(5.84) results(6.80) experiments(7.37) several(2.94) real(1.83) time(1.94) decision(4.34) algorithms(5.16) domain.(8.45) results(10.73) demonstrate(14.65) high(5.51) performance(6.46) two(7.53) algorithms,(6.19) decision(4.79) evaluation(4.18) variant(3.69) incremental(2.25) probabilisitic(2.59) inference(6.22) dambrosio,(3.81) 93(4.42) variant(3.46) algorithm(5.22) suggested(3.74) goldszmidt,(3.28) goldszmidt,(2.38) 95(5.24) ],(3.08) pk(2.77) reduced.(6.20) discuss(13.66) implications(9.36) experimental(9.82) results(9.38) explore(8.79) broader(5.65) applicability(3.44) algorithms.(14.64)
Node-1: title(12.47) learning(12.87) policies(9.77) partially(3.11) observable(2.82) environments(5.58) scaling(9.39) abstract(10.80) partially(4.42) observable(2.62) markov(4.50) decision(5.75) processes(4.53) pomdp(9.69) model(11.47) decision(7.63) problems(7.18) agent(12.00) tries(3.13) maximize(3.05) reward(6.03) face(2.13) limited(2.17) noisy(8.96) sensor(6.27) feedback.(5.17) study(4.64) pomdp(15.31) motivated(4.24) need(2.26) address(2.23) realistic(4.37) problems,(2.86) existing(3.55) techniques(5.16) finding(4.80) optimal(15.47) behavior(5.35) appear(2.47) scale(3.92) well(2.03) unable(3.29) find(3.00) satisfactory(4.62) policies(7.82) problems(4.45) dozen(4.76) states.(10.45) brief(5.22) review(6.37) pomdp(6.59) s,(4.16) paper(10.26) discusses(7.55) several(5.13) simple(4.01) solution(4.55) methods(5.11) shows(4.13) capable(5.08) finding(4.90) near(3.28) optimal(21.93) policies(18.79) selection(5.62) extremely(5.44) small(4.79) pomdp(23.78) taken(4.85) learning(17.53) literature.(7.03) contrast,(4.73) show(3.74) none(6.38) able(2.67) solve(3.53) slightly(3.31) larger(3.38) noisier(3.47) problem(4.47) based(2.61) robot(20.05) navigation.(10.29) find(4.77) combination(3.98) two(3.96) novel(8.04) approaches(5.12) performs(3.93) well(2.61) problems(5.02) suggest(7.78) methods(5.07) scaling(10.48) even(2.50) larger(2.94) complicated(4.88) domains.(8.93) "

---

**Zero-Shot w/ Saliency Prompt Part.2**

**HumanMessage:** "Node-1.1: title(0.95) formal(0.36) framework(0.41) speedup(0.35) learn-ing(0.41) problems(0.48) solutions(0.48) abstract(1.14) speedup(0.33) learning(0.61) seeks(0.57) im-prove(0.27) computational(0.50) efficiency(0.35) problem(0.37) solving(0.41) experience.(0.57) pa-per,(0.70) develop(0.53) formal(0.40) framework(0.38) learning(0.37) efficient(0.40) problem(0.34) solving(0.32) random(0.54) problems(0.32) solutions.(0.37) apply(0.42) framework(0.47) two(0.45) different(0.24) representations(0.54) learned(0.64) knowledge,(0.41) namely(0.58) control(0.99) rules(0.73) macro(0.54) operators,(0.48) prove(0.58) theorems(0.38) identify(0.28) sufficient(0.22) conditions(0.28) learning(0.38) representation.(0.46) proofs(0.50) constructive(0.81) accompa-nied(0.47) learning(0.54) algorithms.(0.65) framework(0.83) captures(1.22) empirical(0.63) expla-nation(0.81) based(0.44) speedup(0.39) learning(0.67) unified(0.80) fashion.(0.54) illustrate(1.50) framework(0.56) implementations(0.79) two(0.46) domains(0.76) symbolic(0.78) integration(0.70) eight(0.65) puzzle.(0.81) work(1.09) integrates(0.74) many(0.54) strands(1.11) experimental(0.75) theoretical(0.55) work(0.49) machine(0.99) learning,(0.66) including(0.61) empirical(0.82) learn-ing(0.59) control(0.85) rules,(0.83) macro(0.81) operator(1.00) learning,(1.31)
Node-1.2: title(2.30) acting(0.98) uncertainty(2.31) discrete(1.03) bayesian(1.13) models(0.94) mo-bile(1.03) robot(1.75) navigation(1.12) abstract(2.80) discrete(1.18) bayesian(0.97) models(0.81) used(0.66) model(0.56) uncertainty(1.79) mobile(0.77) robot(1.55) navigation,(0.66) question(0.64) actions(1.17) chosen(0.67) remains(0.72) largely(0.56) unexplored.(0.74) paper(3.63) presents(1.59) optimal(1.17) solution(1.03) problem,(1.09) formulated(0.96) partially(0.40) observable(0.42) markov(0.90) decision(0.74) process.(0.85) since(1.13) solving(1.17) optimal(1.16) control(0.75) pol-icy(1.02) intractable,(0.74) general,(0.82) goes(1.52) explore(2.04) variety(1.42) heuristic(0.84) con-trol(1.05) strategies.(2.28) control(1.40) strategies(3.63) compared(3.99) experimentally,(1.72) simu-lation(2.58) runs(1.63) robot.(3.17)
Node-1.3: title(1.50) incremental(0.64) methods(0.50) computing(0.59) bounds(1.09) partially(0.24) observable(0.21) markov(0.31) decision(0.64) processes(0.38) abstract(0.97) partially(0.25) ob-servable(0.21) markov(0.32) decision(0.58) processes(0.36) pomdps(0.21) allow(0.54) one(0.36) model(0.47) complex(0.38) dynamic(0.60) decision(0.76) control(0.38) problems(0.53) include(0.31) action(0.82) outcome(0.55) uncertainty(0.61) imperfect(0.54) observabil(0.22) ity.(0.36) control(0.47) problem(0.67) formulated(0.87) dynamic(0.63) optimization(1.30) problem(0.58) value(0.33) func-tion(0.28) combining(0.60) costs(0.97) rewards(0.92) multiple(0.37) steps.(0.53) paper(1.77) propose,(0.72) analyse(0.35) test(0.45) various(0.60) incremental(0.42) methods(0.43) comput-ing(0.52) bounds(0.49) value(0.23) function(0.23) control(0.54) problems(0.62) infinite(0.38) dis-counted(0.33) horizon(0.62) criteria.(0.68) methods(0.60) described(0.81) tested(0.68) include(0.68) novel(1.07) incremental(0.44) versions(0.58) grid(1.00) based(0.30) linear(0.29) interpolation(0.22) method(0.39) simple(0.31) lower(0.28) bound(0.53) method(0.38) sondik(0.34) updates.(0.65) work(0.33) arbitrary(0.38) points(0.47) belief(1.86) space(0.42) enhanced(0.56) various(0.51) heuristic(0.29) point(0.31) selection(0.39) strategies.(0.64) also(0.76) introduced(1.55) new(0.88) method(0.62) computing(0.59) initial(0.46) upper(0.46) bound(0.67) fast(0.43) informed(0.56) bound(0.48) method.(0.56) method(0.44) able(0.39) improve(0.45) significantly(0.33) standard(0.37) commonly(0.37) used(0.21) upper(0.28) bound(0.40) computed(0.73) mdp(0.24) based(0.27) method.(0.41) quality(1.09) resulting(1.01) bounds(1.67) tested(1.72) maze(1.68) navigation(1.34) problem(1.05) 20(0.68) states,(0.88) 6(0.37) actions(1.42) 8(0.69) observations.(1.83)
Node-1.4: title(0.98) learning(1.07) sorting(1.63) decision(1.56) trees(2.00) pomdps(1.04) ab-stract(1.34) pomdps(1.10) general(0.42) models(0.59) sequential(0.99) decisions(0.93) actions(0.63) observations(1.27) probabilistic.(0.59) many(0.37) problems(1.06) interest(0.66) formulated(0.98) pomdps,(1.14) yet(0.44) use(0.34) pomdps(0.54) limited(0.33) lack(0.32) effective(0.49) algo-rithms.(0.73) recently(0.51) started(0.30) change(0.25) number(0.21) problems(0.76) robot(1.39) navigation(0.71) planning(0.60) beginning(0.32) formulated(0.70) solved(0.63) pomdps.(0.59) ad-vantage(0.46) pomdp(0.62) approach(0.80) clean(0.50) semantics(0.85) ability(0.28) produce(0.24) principled(0.33) solutions(0.42) integrate(0.41) physical(0.79) information(0.41) gathering(0.50) actions.(0.71) paper(1.89) pursue(0.91) approach(0.72) context(0.64) two(0.59) learning(0.61) tasks(0.60) learning(0.47) sort(0.46) vector(0.57) numbers(0.48) learning(0.67) decision(2.36) trees(1.73) data.(0.63) problems(1.59) formulated(1.32) pomdps(2.25) solved(0.93) general(0.57) pomdp(1.45) algorithm.(0.82) main(0.56) lessons(1.34) results(0.55) 1(0.48) use(0.39) suitable(0.36) heuristics(0.29) representations(0.49) allows(0.42) solution(0.37) sorting(1.14) classification(0.58) pomdps(0.58) non(0.22) trivial(0.45) sizes,(0.42) 2(0.29) quality(0.31) resulting(0.29) solutions(0.38) competitive(0.37) best(0.32) algorithms,(0.42) 3(0.28) problematic(0.52) aspects(0.42) decision(1.53) tree(1.12) learning(0.63) test(0.45) mis(0.29) classification(0.51) costs,(0.36) noisy(0.76) tests,(0.39) missing(0.32) values(0.36) naturally(0.66) accommodated.(0.88) "

### Zero-Shot w/ Saliency Prompt Part.3

**HumanMessage:** "Node-1.5: title(0.90) approximating(0.42) optimal(0.69) policies(0.65) partially(1.13) observable(0.41) stochastic(0.50) domains(0.63) abstract(1.25) problem(0.51) making(0.22) optimal(0.40) decisions(0.42) uncertain(1.01) conditions(0.80) central(0.56) artificial(0.82) intelligence.(0.75) state(0.54) world(0.72) known(0.29) times,(0.22) world(0.61) modeled(0.35) markov(0.48) decision(0.48) process(0.35) mdp(0.20) ).(0.23) mdps(0.32) studied(0.34) extensively(0.41) many(0.26) methods(0.42) known(0.30) determining(0.35) optimal(0.44) courses(0.42) action,(0.41) policies.(0.70) realistic(0.58) case(0.39) state(0.70) information(0.67) partially(0.43) observable,(0.45) partially(0.70) observable(0.31) markov(0.36) decision(0.52) processes(0.40) pomdps(0.22) ),(0.26) received(0.36) much(0.26) less(0.35) attention.(0.54) best(0.50) exact(0.48) algorithms(0.98) problems(0.75) inefficient(0.25) space(0.28) time.(0.51) introduce(2.21) smooth(0.81) partially(1.31) observable(0.52) value(0.48) approximation(0.99) spova(0.26) ),(0.35) new(1.42) approximation(1.16) method(0.80) quickly(0.52) yield(0.42) good(0.32) approximations(0.40) improve(0.33) time.(0.30) method(0.45) combined(0.42) reinforcement(0.76) learning(0.55) methods,(0.41) combination(0.44) effective(0.48) test(0.63) cases.(0.57)
Node-1.6: title(1.58) efficient(1.08) dynamic(0.83) programming(1.15) updates(2.24) partially(0.70) observable(0.55) markov(0.87) decision(1.19) processes(0.86) abstract(1.67) examine(0.99) problem(0.72) performing(0.50) exact(0.70) dynamic(0.57) programming(0.75) updates(1.48) partially(1.26) observable(0.58) markov(0.78) decision(1.58) processes(0.78) pomdps(1.18) computational(1.04) complexity(0.75) viewpoint.(0.80) dynamic(0.72) programming(0.93) updates(1.74) crucial(0.60) operation(0.47) wide(0.25) range(0.29) pomdp(1.34) solution(0.98) methods(0.64) find(0.43) intractable(0.61) perform(0.44) updates(1.31) piecewise(0.41) linear(0.55) convex(1.90) value(0.88) functions(0.54) general(0.61) pomdps.(1.28) offer(1.36) new(1.15) algorithm,(1.71) called(1.15) witness(7.58) algorithm,(2.37) compute(0.78) updated(1.06) value(0.83) functions(0.60) efficiently(0.83) restricted(0.69) class(0.46) pomdps(0.92) number(0.57) linear(0.66) facets(0.70) great.(1.14) compare(1.60) witness(8.22) algorithm(3.33) existing(1.12) algorithms(1.68) analytically(0.87) empirically(0.88) find(0.95) fastest(1.48) algorithm(1.78) wide(0.44) range(0.44) pomdp(3.93) sizes.(1.29)
Node-2: title(14.46) efficient(7.56) inference(7.77) bayes(5.83) networks(10.92) combinatorial(4.43) optimization(7.56) problem(7.08) abstract(20.68) number(4.43) exact(10.23) algorithms(14.38) developed(3.37) perform(4.58) probabilistic(5.11) inference(6.22) bayesian(9.36) belief(43.68) networks(17.76) recent(7.91) years.(5.88) techniques(5.10) used(3.03) algorithms(12.62) closely(3.05) related(2.74) network(7.05) structures(5.11) easy(2.52) understand(3.09) implement.(8.30) paper,(12.97) consider(9.07) problem(8.09) combinatorial(3.31) optimization(5.31) point(2.81) view(5.13) state(7.40) efficient(7.68) probabilistic(3.59) inference(7.52) belief(36.55) network(12.91) problem(5.46) finding(3.37) optimal(5.77) factoring(4.62) given(3.44) set(2.39) probability(11.09) distributions.(9.15) viewpoint,(7.21) previously(5.96) developed(4.64) algorithms(8.95) seen(6.21) alternate(6.64) factoring(3.80) strategies.(8.92) paper,(9.32) define(20.14) combinatorial(10.03) optimization(13.81) problem,(10.52) optimal(13.04) factoring(4.44) problem,(5.48) discuss(12.35) application(5.49) problem(8.90) belief(42.48) networks.(16.00) show(9.32) optimal(7.28) factoring(4.10) provides(4.40) insight(7.52) key(3.28) elements(3.51) efficient(10.67) probabilistic(5.69) inference,(7.93) demonstrate(13.60) simple,(4.61) easily(2.91) implemented(3.21) algorithms(9.27) excellent(6.64) performance.(9.49)
Node-2.1: title(0.96) sensitivities(0.32) alternative(0.73) conditional(0.51) probabilities(0.29) bayesian(0.53) belief(1.27) networks(0.93) abstract(1.29) show(0.82) alternative(0.44) way(0.32) representing(0.60) bayesian(0.48) belief(1.94) network(1.51) sensitivities(0.25) probability(0.69) distributions.(0.55) representation(0.53) equivalent(0.34) traditional(1.06) representation(0.54) conditional(0.36) probabilities,(0.26) makes(0.24) dependencies(0.29) nodes(1.00) apparent(0.68) intuitively(0.28) easy(0.28) understand.(0.69) also(0.53) propose(1.01) qr(0.28) matrix(0.71) representation(0.65) sensitivities(0.26) conditional(0.31) probabilities(0.21) efficient,(0.25) memory(0.44) requirements(0.26) computational(0.35) speed,(0.26) traditional(0.64) representation(0.53) computer(0.52) based(0.21) implementations(0.49) probabilistic(0.21) inference.(0.62) use(0.63) sensitivities(0.51) show(1.23) certain(0.44) class(0.42) binary(0.73) networks,(0.74) computation(0.50) time(0.32) approximate(0.64) probabilistic(0.27) inference(0.62) positive(0.23) upper(0.31) bound(0.30) error(0.45) result(0.25) independent(0.32) size(0.26) network.(0.85) finally,(1.12) alternative(0.86) traditional(0.81) algorithms(1.18) use(0.44) conditional(0.79) probabilities,(0.42) describe(1.47) exact(1.02) algorithm(1.54) probabilistic(0.25) inference(0.98) uses(0.48) qr(0.27) representation(0.54) sensitivities(0.45) updates(1.16) probability(0.89) distributions(0.74) nodes(0.93) network(1.40) according(0.75) messages(1.33) neigh(1.47) bors.(0.88) "

**Zero-Shot w/ Saliency Prompt Part.4**

**HumanMessage:** "Node-2.2: title(1.81) algebraic(1.21) techniques(0.51) efficient(0.73) inference(0.70) bayesian(0.55) networks(1.04) abstract(1.57) number(0.37) exact(0.77) algorithms(1.75) developed(0.34) perform(0.43) probabilistic(0.35) inference(0.59) bayesian(0.50) belief(1.91) networks(1.06) recent(1.20) years.(0.81) algorithms(0.79) use(0.32) graph(0.60) theoretic(0.24) techniques(0.43) analyze(0.40) exploit(0.42) network(0.50) topology.(0.79) paper,(1.07) examine(1.16) problem(0.52) efficient(0.56) probabilistic(0.25) inference(0.57) belief(1.60) network(0.88) combinatorial(0.29) optimization(1.34) problem,(0.45) finding(0.35) optimal(0.55) factoring(0.70) given(0.46) algebraic(0.78) expression(0.59) set(0.20) probability(0.69) distributions.(1.04) define(1.24) combinatorial(0.33) optimization(1.91) problem,(0.57) optimal(0.55) factoring(0.41) problem,(0.40) discuss(1.22) application(0.49) problem(0.46) belief(1.47) networks.(1.33) show(0.80) optimal(0.56) factoring(0.42) provides(0.29) insight(0.47) key(0.24) elements(0.31) efficient(0.89) probabilistic(0.32) inference,(0.53) present(0.61) simple,(0.35) easily(0.28) implemented(0.31) algorithms(0.82) excellent(0.41) performance.(0.61) also(0.64) show(1.07) use(0.41) algebraic(0.88) perspective(0.65) permits(0.76) significant(0.52) extension(0.70) belief(2.80) net(1.18) representation.(1.00)
Node-2.3: title(1.43) interpretation(0.76) complex(0.42) scenes(0.80) using(0.46) bayesian(0.87) networks(0.89) abstract(1.12) object(0.85) recognition(1.42) systems,(0.46) interactions(0.45) objects(0.43) scene(0.56) ignored(0.38) best(0.26) interpretation(0.69) considered(0.24) set(0.22) hypothesized(0.20) objects(0.39) matches(0.23) greatest(0.26) number(0.20) image(0.50) features.(0.79) show(0.97) image(0.67) interpretation(1.10) cast(0.81) problem(0.77) finding(0.41) probable(0.63) explanation(0.60) mpe(0.22) bayesian(0.69) network(0.67) models(0.30) visual(0.64) physical(0.45) object(0.71) interactions.(0.83) problem(0.59) determine(0.39) exact(0.40) conditional(1.18) probabilities(0.62) network(1.07) shown(0.68) unimportant,(0.35) since(0.29) goal(0.42) find(0.27) probable(0.47) configuration(0.62) objects,(0.48) calculate(0.56) absolute(0.44) probabilities.(0.50) furthermore(1.36) show(0.88) evaluating(0.71) configurations(1.15) feature(1.08) counting(0.49) equivalent(0.59) calculating(0.52) joint(0.44) probability(0.57) configuration(0.93) using(0.42) restricted(0.62) bayesian(0.57) network,(0.48) derive(1.14) assumptions(0.95) probabilities(0.38) necessary(0.30) make(0.23) bayesian(0.66) formulation(0.85) reasonable.(0.80)
Node-2.4: title(1.04) case(0.45) based(0.40) probability(0.95) factoring(0.36) bayesian(0.53) belief(1.33) networks(1.03) abstract(1.04) bayesian(0.90) network(0.97) inference(0.87) formulated(0.55) combinatorial(0.30) optimization(0.73) problem,(0.41) concerning(0.65) computation(0.55) optimal(0.54) factoring(0.40) distribution(1.54) represented(0.70) net.(2.87) since(0.28) determination(0.37) optimal(0.35) factoring(0.24) computationally(0.29) hard(0.29) problem,(0.32) heuristic(0.33) greedy(1.22) strategies(0.44) able(0.25) find(0.18) approximations(0.31) optimal(0.36) factoring(0.24) usually(0.31) adopted.(0.54) present(0.98) paper(1.04) investigate(0.86) alternative(0.66) approach(0.77) based(0.53) combination(0.44) genetic(1.07) algorithms(0.92) ga(0.80) case(0.49) based(0.74) reasoning(2.04) cbr(0.46) ).(1.16) show(0.69) use(0.44) genetic(1.01) algorithms(0.80) improve(0.37) quality(0.32) computed(0.71) factoring(0.45) case(0.37) static(0.66) strategy(0.44) used(0.25) mpe(0.28) computation(0.74) ),(0.39) combination(0.50) ga(1.10) cbr(0.60) still(0.33) provide(0.35) advantages(0.58) case(0.34) dynamic(0.64) strategies.(0.77) preliminary(1.21) results(0.84) different(0.69) kinds(0.77) nets(3.99) reported.(1.79)
</verbalized-graph>
### Classification Label Probabilistic Methods
### Reasoning
0. Graph Structure Reconstruction:
In the provided verbalized graph, The ROOT node (first line) is the target for classification.
Single-digit indexed nodes are direct neighbors of ROOT.
Double-digit indexed nodes are:
- Two hops away from ROOT
- Direct children of their parent node
More digits indexed nodes follow the same principle as described above."

---

**Zero-Shot w/ Saliency Prompt Part.5**

**HumanMessage:** "Thus, the graph structure of this verbalized graph is:

- ROOT
    - Node-1
        * Node-1.1
        * Node-1.2
        * Node-1.3
        * Node-1.4
        * Node-1.5
        * Node-1.6
    - Node-2
        * Node-2.1
        * Node-2.2
        * Node-2.3
        * Node-2.4

1. Word-Level Evaluation:

Detect important terms for the classification label.
Quantitatively, the importance (saliency) scores behind each word in the verbalized graph are calculated by a post-hoc XAI (explainable AI) algorithm.
Semantically, keywords synonymous with or closely related to label, 'Probabilistic Methods' (such as 'probabilistic', 'probability', and 'inference'), are semantically influential in determining the classification because they directly indicate the node's relevance to the 'Probabilistic Methods' category.
2. Graph-Level Aggregation:

Node 1 does not contain as many directly relevant keywords. Yet, it still discusses POMDPs, which are probabilistic models, indicating that it contributes to the classification in a less significant way.
Direct neighbors like Node 2, which contains the keyword 'probabilistic', have a substantial influence on ROOT node classification due to their close proximity to the ROOT node.
Indirect neighbors, Nodes-2.1 to Node-2.4, connected through Node 2, contain keywords like 'probabilistic' and 'probability', which further reinforce the classification of ROOT.
Meanwhile, the indirect neighbors connected through Node 1, Node-1.1 to Node-1.6, are less relevant because they do not include important keywords directly related to 'Probabilistic Methods'.

### Free-Text Explanation "'markdown The classification of ROOT node into the "Probabilistic Methods" category can be explained as follows:
ROOT: The presence of keywords like "probabilistic" and "inference" within ROOT node suggests that the paper is focused on topics central to probabilistic methods, such as performing probabilistic inference and evaluating influence diagrams.
- Node-1: While Node 1 focuses on POMDPs, which are also probabilistic models, it is slightly less directly related to the probabilistic inference methods emphasized in ROOT node.
- Node-1.1   Node-1.6: they are less relevant to probabilistic methods and thus contribute less to the classification.
- Node-2: This node discusses "probabilistic inference" and "Bayesian networks," both of which are foundational to probabilistic methods. The strong thematic connection between ROOT node and Node 2 reinforces the classification.
- Node-2.1   Node-2.4: they also discuss probabilistic topics, further supporting the classification.

In summary, the classification of ROOT node into "Probabilistic Methods" is well-supported by the strong presence of key probabilistic terms within the node itself and its direct connection to Node 2, which also focuses on probabilistic inference. Indirect neighbors through Node 2 further reinforce this classification."

---

**Zero-Shot w/ Saliency Prompt Part.6**

**HumanMessage:** "## Task
### Verbalized Graph
`<verbalized-graph>`{document}`</verbalized-graph>`
### Classification Label
`{label}`
### Reasoning
### Free-Text Explanation
*(P.S.: 1. make sure to complete both the reasoning section and then Free-Text Explanation section with the same structure as exemplified above.*
*2. make good use of the importance (saliency) score behind each word as your guidance to generate the better explanation. However, it is not necessary to directly quote the saliency score.*
*3. use the **whole** graph structure you constructed during reasoning for the format of the explanation. Indents and node indexes are necessary, which represent the hierarchy of the graph.)"*

---

