# OpenReview forum: "TAGExplainer: Narrating Graph Explanations for Text-Attributed Graph Learning Models"
_ICLR.cc/2025/Conference — ICLR 2025 Conference Withdrawn Submission_

### Official Review · Reviewer_hKeJ · 2024-11-03

**Soundness:** 3
**Presentation:** 3
**Contribution:** 3
**Rating:** 5
**Confidence:** 3

**Summary:**

The paper introduces a novel explanation generation framework, *TAGExplainer*, the first natural language explanation generation method explicitly designed for TAG learning. By generating pseudo-labels and integrating expert iterative training, TAGExplainer effectively enhances the faithfulness and brevity of the explanations. Experimental results show that TAGExplainer outperforms existing models on various datasets, including Cora, DBLP, and Book-History. Its key innovation lies in its model-agnostic design, making it applicable to multiple TAG models while producing explanations that are more human-friendly.

**Strengths:**

- **Originality**: This paper pioneers an innovative framework to generate natural language explanations for TAG learning models. Building upon the existing saliency-based TAG explanation approaches, it combines importance-ranking BFS tree construction and pre-order traversal to generate *Saliency Paragraphs*, and uses a pseudo-label generator to address the lack of labeled data in practical applications. The generated pseudo-labels train the explanation model, making the framework adaptable to various TAG models while yielding more human-centric explanations.

- **Quality**: The experimental design is comprehensive, covering three commonly used datasets in the graph representation learning domain and comparing several mainstream LLMs as baselines. Results demonstrate that TAGExplainer significantly outperforms other baselines in faithfulness and brevity, especially advanced models like GPT-4.

- **Clarity**: The paper is well-structured, with clear problem statements and motivations. Each component of the TAGExplainer framework is illustrated with diagrams, several clear examples, and complete prompts.

- **Significance**: TAGExplainer provides TAG learning models with a new natural language explanation generation method, significantly improving decision interpretability. Compared to existing saliency-based TAG explanations, the generated explanations excel in readability and conciseness.

**Weaknesses:**

- **Pseudo-label Quality**: The generation of pseudo-labels relies on LLMs, yet the high fidelity between input and output does not directly confirm the quality of pseudo-labels. Additional evaluation metrics and methods are required to comprehensively assess pseudo-label quality, potentially yielding better results with LLM-based assessments.

- **Simplicity of Brevity Metric**: The brevity metric may be too simplistic; for example, using the average length ratio of the explanation to the input length as a brevity metric might be inadequate, as explanations should not be excessively short. Overly concise explanations may lack sufficient information, leading to misunderstanding.

-  **Potential Bias in Filtering and Evaluation**: Using the same metric for both filtering and evaluating quality may reduce the credibility of the results. Employing independent evaluation criteria would help confirm the effectiveness of the filtering process and avoid subjective interference.

- **Figure Improvement**: The red markers in Figure 2 are only clearly explained in Figure 4. Improving clarity here would enhance readability.

**Questions:**

- The authors propose a graph-to-text conversion method that uses importance-ranked BFS tree construction and pre-order traversal, claiming that this method preserves structural, semantic, or feature importance information. Could this be demonstrated through experiments or examples?

- The authors mention that the motivation behind generating pseudo-labels is due to the difficulty in obtaining explanation labels, but why not construct a small sample of real labels as examples?

-  Are explanations generated purely by LLMs and filtered with simple metrics high enough in quality?

- Could human evaluation be used to validate the rationality and effectiveness of the metrics?

---

### Official Review · Reviewer_As9U · 2024-11-04

**Soundness:** 2
**Presentation:** 3
**Contribution:** 2
**Rating:** 3
**Confidence:** 3

**Summary:**

This paper proposes to use LLMs to provide natural language explanations for TAG models, as an alternative to the original saliency-based explanations.

**Strengths:**

1. Textual explanations are stronger than saliency-based explanations.
2. The explanation algorithm provided in this paper is model-agnostic.
3. The explanation algorithm has advantages in metrics.

**Weaknesses:**

1. Overall, I believe that this paper does not address a core academic problem. The paper merely transforms saliency-based explanations into text. However, it does not provide information beyond saliency. While text can indeed improve the readability of explanations, this does not seem to reach the level of [providing additional insights]
2. I believe that providing textual explanations for TAG models is straightforward. The experiments in this paper seems confirm this point. For example, as shown in Table 1, direct use of GPT-4o or GPT-3.5 also achieved good results. If direct use of GPT-4o can achieve significant effects, what is the purpose of proposing such a complex TAGExplainer?
3. The method proposed in this paper merely transforms saliency-based explanations into text. So, what new explanatory information does this approach provide compared to saliency-based explanations? The example in Figure 4 seems somewhat arbitrary.
4. From table 2, it appears that the effect brought by different modules of TAGExplainer is not significant. This raises questions about the motivation for designing such a complex architecture.
5. According to figure 2, it seems that the TAG expert Iteration and the graph are unrelated. If so, why did this paper specifically choose graph instead of a more general saliency-based explanation?

**Questions:**

Please see weaknesses.

---

### Official Review · Reviewer_Bcc7 · 2024-11-04

**Soundness:** 3
**Presentation:** 3
**Contribution:** 2
**Rating:** 5
**Confidence:** 3

**Summary:**

This paper introduces TAGExplainer, a two-step framework for generating natural language explanations for Text-Attributed Graph (TAG) learning methods. TAGExplainer enhances any saliency-based TAG explanation approach by first verbalizing the Saliency Textual Graph. Using a pre-order traversal on the BFS, TAGExplainer arranges the graph in a structured format that reflects the original graph’s layout, placing more important nodes at the beginning to emphasize their significance, and explicitly add the importance score after each token.

TAGExplainer is optimized using Expert Iteration, selecting the high-quality explanations using the combination of the following metrics: 1) point-wise mutual information (PMI) between explanation and true rationale (important nodes/tokens), 2) PMI between explanation and model's prediction, 3) brevity. Finally, an end-to-end student explainer that only requires the initial graph is trained based on the previous teacher model that requires the saliency-based TAG explanation. Experiments on 3 real world datasets shows the effectiveness of the proposed TAGExplainer that can balance both faithfulness and brevity compared to directly querying large, general-purpose LLMs like GPT-4o.

**Strengths:**

1. The problem of generating human readable explanations for TAG explainers is important.
2. The overall methodology is simple but effective and can be used for any saliency-based TAG explainers.
3. The training of expert iteration only requires a handful high-quality samples.

**Weaknesses:**

1. Some aspects are not clear, see the question section.

**Questions:**

1. The explanation of faithfulness calculation in Section 4.2.1 lacks clarity. For instance, how is the gemma model used to perform the MLM task in Eq.2 (maybe using perplexity here)? Additionally, since tokens are sampled to construct $R_\tau$, how is the graph structure maintained in both $R_\tau$ and $S_{M_r}$? Providing a detailed example of how Eq.2 is computed would enhance understanding. Similarly, there is a question regarding the computation of Eq.3. Shouldn’t it also be conditioned on the input subgraph to allow rewriting it to be $\log \frac{P(\hat y |S,E)}{P(\hat y|S)}$?
2. The authors mentioned that "we utilized the fine-tuned gemma2-2b-it model to estimate the conditional probability distribution". May I ask how exactly the fine-tuning is performed here?
3.  When constructing the verbalization of the saliency textual graph, nodes are re-ranked based on their node-level importance score in order to provide "implicitly" saliency information to the LLM. Does this design contribute significantly to the final results?

---

### Note · Authors · 2024-12-16

**Comment:**

We appreciate all the reviewers for their important and insightful comments. After careful consideration, we decide to withdraw this submission.

**Withdrawal Confirmation:**

I have read and agree with the venue's withdrawal policy on behalf of myself and my co-authors.